# Effects of Partial Manganese Substitution by Cobalt on the Physical Properties of Pr_0.7_Sr_0.3_Mn_(1−x)_Co_x_O_3_ (0 ≤ x ≤ 0.15) Manganites

**DOI:** 10.3390/ma16041573

**Published:** 2023-02-13

**Authors:** Feriel Zdiri, José María Alonso, Taoufik Mnasri, Patricia de la Presa, Irene Morales, José Luis Martínez, Rached Ben Younes, Pilar Marin

**Affiliations:** 1Laboratory of Technology, Energy and Innovative Materials, Department of Physics, Faculty of Sciences of Gafsa, Gafsa 2112, Tunisia; 2Instituto de Magnetismo Aplicado, Universidad Complutense de Madrid, A6 22,500 Km, Las Rozas, 28230 Madrid, Spain; 3Instituto de Ciencia de Materiales, CSIC, Sor Juana Inés de la Cruz s/n, 28049 Madrid, Spain; 4Departamento de Física de Materiales, Universidad Complutense de Madrid, 28040 Madrid, Spain

**Keywords:** Mn-perovskite, X-ray diffraction, magnetization, resistivity and electrical conduction, magnetoresistance (MR)

## Abstract

We have investigated the structural, magnetic, and electrical transport properties of Pr_0.7_ Sr_0.3_ Mn_(1−x)_Co_x_ O_3_ nanopowders (x = 0, 0.05, 0.10 and 0.15). The Pechini Sol-gel method was used to synthesize these nanopowders. X-ray diffraction at room temperature shows that all the nano powders have an orthorhombic structure of Pnma space group crystallography. The average crystallite size of samples x = 0, 0.05, 0.10, and 0.15 are 33.78 nm, 29 nm, 33.61 nm, and 24.27 nm, respectively. Semi-quantitative chemical analysis by energy dispersive spectroscopy (EDS) confirms the expected stoichiometry of the sample. Magnetic measurements indicate that all samples show a ferromagnetic (FM) to paramagnetic (PM) transition with increasing temperature. The Curie temperature T_C_ gradually decreases (300 K, 270 K, 250 K, and 235 K for x = 0, 0.05, 0.10, and 0.15, respectively) with increasing Co concentrations. The M-H curves for all compounds reveal the PM behavior at 300 K, while the FM behavior characterizes the magnetic hysteresis at low temperature (5 K). The electrical resistivity measurements show that all compounds exhibit metallic behavior at low temperature (T < Tρ) well fitted by the relation ρ = ρ_0_ + ρ_2_T^2^ + ρ_4.5_T^4.5^ and semiconductor behavior above Tρ (T > Tρ), for which the electronic transport can be explained by the variable range hopping model and the adiabatic small polaron hopping model. All samples have significant magnetoresistance (MR) values, even at room temperature. This presented research provides an innovative and practical approach to develop materials in several technological areas, such as ultra-high density magnetic recording and magneto resistive sensors.

## 1. Introduction

During the past few years, the Mn based perovskite oxides have attracted the scientific community due to their properties, such as electronic transport, magnetic properties, and colossal magneto resistance (CMR) [1,2,3] near their paramagnetic-ferromagnetic (PM-FM) transition temperature T_C_, which is known as one of the future resources for improving the performance of microelectronic devices, especially those that detect and record magnetic signals. In the structure of perovskite manganites $({\mathrm{Tr}}_{1-x}^{3+}A_{X}^{2+})\;\left({\mathrm{Mn}}_{1-x}^{3+}{\mathrm{Mn}}_{x}^{4+}\right)O_{3}^{2-}$, where (Tr^3+^ = La^3+^, Sm^3+^,Pr^3+^, Nd^3+^, etc.., and A^2+^ = Ca^2+^, Ba^2+^, Sr^2+^, etc..), the substitution of trivalent rare earth ions (Tr^3+^) by a divalent element (A^2+^) in the perovskite structure is one way of varying their physical properties. The partial substitution of Tr^3+^ leads to the appearance of a mixed valence of manganese Mn^3+^ and Mn^4+^. This valence state is at the base of the changes in the physical properties of these manganites, in particular the appearance of a ferromagnetic order of the spins of the manganese ions, following which the electron e_g_ becomes itinerant and can hop from a manganese cation, via the oxygen anion, to another manganese having a completely empty e_g_ band. One of the fundamental characteristics of manganites is the high correlation between structure, electrical properties, and magnetic properties. This correlation has been explained by the double exchange (DE) mechanism introduced by Zener in 1951 [4]. Previous studies [5,6,7] have shown that electron–phonon interactions in the form of Jahn–Teller polarons, orbital order, electron-electron correlations, and coupling between spin and orbital structures can also play a very important role. These interactions are affected by several intrinsic parameters, such as the chemical and electronic nature of the A and B atoms [8,9], oxygen stoichiometry [10,11], substitution rate [12], particle size [13], etc. Recently, numerous scientific researches [14,15,16] on manganites have shown that the substitution of rare earths indirectly affects the conduction mechanism, with an impact on the bandwidth and angle of the bond between neighboring manganese ions. There is also no doubt that Mn ions play a major role in the double exchange interaction, so it is interesting to study the influences of the substitution of Mn by other elements. Several studies [17,18,19,20,21] have been carried out over the last few decades to understand the effect of the substitution of a transition element for manganese at the B-site. It has been shown that the introduction of a transition metal with an electronic configuration different from that of Mn should lead to dramatic effects related to this change in the configuration of Mn and that of the substituent element. Here, the substitution of Mn directly affects the conduction mechanism, allowing the physical properties of the manganite system to be adapted more efficiently. In addition, the substitution of Mn by trivalent and tetravalent elements leads to a decrease in transition temperature (T_C_) and magnetization, an increase in resistivity. However, the exact effect depends mainly on the nature of the substituted elements. Among these elements, Cobalt has an effect on the physical properties of manganites. Substitutions of Mn by magnetic Co have been carried out in the systems ${\;\mathrm{La}}_{0.67}{\mathrm{Pb}}_{0.33}{\mathrm{Mn}}_{\left(1-x\right)}{\mathrm{Co}}_{x}O_{3}$ [22,23],${\;\mathrm{La}}_{0.7}{\mathrm{Pb}}_{0.3}{\mathrm{Mn}}_{\left(1-x\right)}{\mathrm{Co}}_{x}O_{3}$ [24], ${\mathrm{La}}_{0.8}{\mathrm{Ba}}_{0.1}{\mathrm{Mn}}_{\left(1-x\right)}{\mathrm{Co}}_{x}O_{3}$ [25], and ${\mathrm{La}}_{0.67}{\mathrm{Sr}}_{0.33}{\mathrm{Mn}}_{\left(1-x\right)}{\mathrm{Co}}_{x}O_{3}$ [26]. According to these studies, the presence of cobalt in these systems leads to similar results, in particular a decrease in the temperature of the magnetic transition from the FM state to the PM state when the rate of substitution x increases. In the ${\mathrm{Pr}}_{0.7}{\mathrm{Ca}}_{0.3}{\mathrm{Mn}}_{\left(1-x\right)}{\mathrm{Co}}_{x}O_{3}$ system, A. Selmi et al. [27] found that cobalt increases the magnetic transition temperature T_C_. While studies by I.G. Diack et al. [28] in the ${\mathrm{Pr}}_{0.7}{\mathrm{Ca}}_{0.3}{\mathrm{CoO}}_{3\;}$ and ${\mathrm{Pr}}_{0.7}{\mathrm{Sr}}_{0.3}{\mathrm{CoO}}_{3}$ systems have shown that T_C_ decreases with increasing cobalt content. In order to widen the field of investigation, especially the search for materials with possibility to be used in magnetic or electrical devices, we studied the properties of some manganites derived from the compound Pr_0.7_Sr_0.3_MnO_3_ [29], whose lanthanide (or rare earth) is praseodymium, much less studied than its counterpart lanthanum. Thus, the structure of Pr_1−x_Sr_x_MnO_3_ can be derived from cubic perovskite by tilting all oxygen octahedrons around [110] pseudo-cubic axes (tilting system a^-^a^-^c^+^). In this paper, we report the structural, thermal, magnetic, and transport properties of ${\mathrm{Pr}}_{0.7}{\mathrm{Sr}}_{0.3}{\mathrm{Mn}}_{\left(1-x\right)}{\mathrm{Co}}_{x}O_{3}$ (0 ≤ x ≤ 0.15) prepared by the Pechini Sol-gel method to shed light on the role of the Co element in the Mn sub lattice.

## 2. Materials and Methods

### 2.1. Experimental Details

Nanocrystalline samples of ${\mathrm{Pr}}_{0.7}{\mathrm{Sr}}_{0.3}{\mathrm{Mn}}_{\left(1-x\right)}{\mathrm{Co}}_{x}O_{3}$ (0 ≤ x ≤ 0.15) were prepared by the Pechini Sol-gel method [30] using stoichiometric amounts of Pr(NO)_3_.6H_2_O, Sr(NO_3_)_2_, Mn(NO_3_)_2_.4H_2_O, and Co(NO_3_).6H_2_O with purity greater than 99.9% (Sigma-Aldrich, St. Louis, MI, USA). The first step is to dissolve a certain amount of carboxylic acids such as citric acid (C_6_H_8_O_7_.H_2_O) (99.5% purity) directly in an aqueous medium (distilled water) to form stable solutions of metal complexes. As soon as the solution becomes homogeneous, the metal nitrates are added one by one. Then, by adding compounds comprising hydroxyl groups, such as ethylene glycol (C_2_H_6_O_2_) (99% purity), a polymeric resin will be able to form upon evaporation of the solvent by polyesterification, and the solution is heated on hot plate with regular stirring for some time at a moderate temperature of 60 °C, producing a homogeneous solution. The resulting solution was dried at 70 °C–85 °C to promote polymerization and allow removal of the solvent. A viscous gel was formed and heated to 100 °C. This step was performed to ensure the propagation of the combustion that transformed the gel into a powder. When the excess solvent is removed by heating, no crystallization or segregation occurs. A solid resin containing homogeneously distributed metal ions is produced. This resin is heated to 500 °C for 24 h to remove organic residues. A black powder is obtained, which is then ground into granules and sintered at 950 °C for 24 h in air to ensure good crystallization of the material.

### 2.2. Characterizations

Thermogravimetric analysis (TGA) was performed in the temperature range of 0–950 °C using a CAHN-D200 electro balance (located at the IMA-UCM-ADIF in Madrid—Spain) with a sensitivity of 10^−7^g at a heating rate of 6 °C/min for 6 h, a reducing atmosphere of 200 mbar of H_2_ and 300 mbar of He, in order to determine and control the oxygen content of the materials with a precision of 0.002 (±2) in its stoichiometry. This content determines the oxidation states of the cations which constitute them, their crystallographic structure, the magnetic order, and thus several of their electronic properties. Structural characterization was performed by X-ray diffraction (XRD) to confirm the crystallinity, purity, and single-phase formation of the samples using a PANalytical X’Pert Pro diffractometer with CuKα radiation (λ = 1.5406Å, Ratio K-Alpha2/K- Alpha1 = 0.5) in a range of 2θ from 10° to 90° with a step size of 0.04 and a counting time of 0.39 s per step. Rietveld structure refinement was used with Fullprof software (version: April 2019) [31] to estimate lattice parameters, space groups, crystal system type, Bragg reflections, and other sample statics. The microstructure and surface morphology of the samples were studied by scanning electron microscopy (SEM) on a JEOL 6400 JSM operating at 25 KV. The transmission electron microscope (TEM), JEOL JEM 2100 operating at 200 KV, located at the CNME in Madrid, Spain was used to determine the average size, shape, and nature of the particle size distribution. For the determination of mean size and distribution, at least 130 particles were analyzed by measuring their longest length using Image J -Win 32 software (version 1.46) and fitted to a log normal distribution. The chemical composition of the sample was studied by energy dispersive spectroscopy (EDS) coupled to TEM. Magnetic measurements of the samples were performed under zero field cooling (ZFC) and field cooling (FC) conditions with an applied field of 0.05T between 5 K and 350 K using a SQUID magnetometer (Quantum Design, MPMS-XL). The isothermal magnetization curves M (H, T) were recorded at 5 K and 300 K, with increasing and decreasing applied fields between 0 and 5 T. The resistivity measurements were performed on a Physical Properties Measurement System (PPMS) from Quantum Design, in a temperature range from 5 to 380 K and with an external magnetic field up to 14 T. The nanopowders were pressed into pellets 13 mm in diameter and 1 mm thickness under 7 Tons/cm^2^.

## 3. Results and Discussion

### 3.1. Thermogravimetric Analysis

In order to determine the oxygen content, the compound ${\mathrm{Pr}}_{0.7}{\mathrm{Sr}}_{0.3}{\mathrm{Mn}}_{\left(1-x\right)}{\mathrm{Co}}_{x}O_{3}$ with (0 ≤ x ≤ 0.15) was analyzed by thermogravimetry (TGA). Figure 1 shows the TGA curves of the prepared samples.

At room temperature to 300 °C the samples undergo a slight weight loss (~0.20%) which corresponds to the removal of water and residual organic matter from the Sol-Gel process [32,33]. From 300 °C to 600 °C, samples undergo a rapid process, losing about 5% of their initial mass accompanied by different thermal events, which corresponds to the weight loss associated with the reduction reaction of the material (Equation (1)):Pr_0.7_Sr_0.3_Mn_1−x_Co_x_O_3_→(0.7/2)·Pr_2_O_3_ + 0.3·SrO + (1−x)·MnO + x·Co(1)

Finally, after 600 °C, no significant weight loss was observed. A tight similarity was observed in the four profiles of the samples carried out by CAHN D-200, such that the oxygen content is equal to 3 (Table 1).

### 3.2. Structural and Morphological Characterization

#### 3.2.1. X-ray Diffraction Results

X-ray diffraction patterns of crystalline samples of nominal formula ${\mathrm{Pr}}_{0.7}{\mathrm{Sr}}_{0.3}{\mathrm{Mn}}_{\left(1-x\right)}{\mathrm{Co}}_{x}O_{3}$ with (0 ≤ x ≤ 0.15) sintered at 950 °C are shown in Figure 2a. We note that all samples show reflections typical of the perovskite structure with Orthorhombic symmetry in the Pnma space group. This was confirmed by the Goldschmidt T_G_ tolerance factor (Equation (2)) in order to verify the existence of perovskite type structures in these solid solutions of nominal formula ${\mathrm{Pr}}_{0.7}^{3+}{\mathrm{Sr}}_{0.3}^{2+}{\mathrm{Mn}}_{0.7-x}^{3+}{\mathrm{Mn}}_{0.3}^{4+}{\mathrm{Co}}_{x}^{3+}O_{3}^{2-}$ taking into account the charge balance [34].
(2)$$t_{G}=\frac{\langle r_{A}\rangle +r_{O}}{\sqrt{2}\;\left(\langle r_{B}\rangle +r_{O}\right)}$$
where r_A_, r_B_, and *r*_O_ are, respectively, the ionic radii of the A, B, and oxygen sites of the ABO_3_ perovskite.

Here,
<r_A_> = 0.7 r_(Pr_^3+^_)_ + 0.3 r_(Sr_^2+^_)_, <r_B_> = (0.7 − x) r_(Mn_^3+^_)_ + 0.3 r_(Mn_^4+^_)_ + x r_(Co_^3+^_)._
<*r*_O_> = r_(O_^2−^_)_ = 1.41 Å, r_(Pr_^3+^_)_ = 1.179 Å, r_(Sr_^2+^_)_ = 1.31 Å, r_(Mn_^3+^_)_ = 0.645 Å, r_(Mn_^4+^_)_ = 0.53 Å, r_(Co_^3+^_)_ = 0.61 Å.

The values of the ionic radii of the different cations were taken from the Shannon studies [35]. Since orthorhombic distortion is allowed for $0.75\leq T_{G}\leq 0.95$, the T_G_ values obtained in Table 2 confirm that our compounds crystallize in this system.

Sharp peaks are clearly visible in all XRD patterns, sign of a good crystallization of the synthesized samples. The structural refinement of these samples was performed with the Fullprof program using the Rietveld powder diffraction technique [36]. This technique is based on a polynomial function, a pseudo-Voigt function, that was used to model the shape of the peak. The results of the refinement obtained are classified in the Table 2. In this table, the factors of relatability are also indicated: Rp, Rwp are factors relating to the profile of the peaks, χ^2^ the goodness of fit, and R_F_ is a factor of confidence relating to the factors of structure. These parameters confirm that there is good agreement between the calculated and observed data and indicate that the refinements are acceptable and that the compositions of the samples are the same as their nominal compositions, including that the oxygen content is close to 3 for all samples. From these results, it can be deduced that the partial substitution of manganese (Mn) ions by cobalt (Co) ions leads to the decrease of the lattice parameters, in particular the elementary lattice volume varying from V = 229.738 Å^3^ for x = 0 to V = 228.878 Å^3^ for x = 0.15. This decrease can be justified by the diffraction peaks in the inset of Figure 2a where these peaks shifted slightly to higher values of 2θ (32.847°; 32.816°, 32.851° and 32.889°) as the substitution rate of cobalt x increases. This can be explained by the fact that the ionic radius of Co^3+^ (r_Co3+_ = 0.61Å) is slightly smaller than that of the Mn^3+^ ion (r_Mn_^3+^ = 0.645 Å), indicating the successful replacement of Co^3+^ at the Mn site in the lattice structure. However, it is much larger at Mn^4+^ (r_Co3+_ = 0.61 Å > r_Mn4+_ = 0.53 Å), justifying the increase in bond length <d_Mn,Co-O_> and decrease in bond angle <θ_Mn,Co-O-Mn,Co_> (These data were obtained using VESTA software (version 3.5.8) [37]). An increase in the Mn,Co-O bond length with an increase in Co content leads to a decrease in the bandwidth W [38] (see Table 2 and Figure 3), which affects the magnetic properties of manganite, and this will be discussed in the following paragraphs.

The average crystallite size of the samples (Table 3) has been estimated from the XRD data using the Scherrer formula [39]:(3)$$D_{\mathrm{SCH}}=\frac{K\;\lambda }{\beta \cos\left(\theta \right)}$$
where K is constant (K = 0.9), λ is the X-ray wavelength (CuKα radiation, λ = 1.5406 Å), β is the experimental full width at half-maximum (FWHM), and θ is the diffraction angle of the most intense peak (hkl) ≡ (121) for all samples.

#### 3.2.2. Morphological Characterization

The structural morphology of the samples ${\mathrm{Pr}}_{0.7}{\mathrm{Sr}}_{0.3}{\mathrm{Mn}}_{\left(1-x\right)}{\mathrm{Co}}_{x}O_{3}$ sintered at 950 °C for x = 0, 0.05, 0.10, and 0.15 was studied using SEM. The typical SEM micrographs are shown in Figure 4. These micrographs show that all samples are tightly packed in quasi-spherical shapes with a fine granulation structure. All the images are almost similar, with no chemical contrast between the crystalline grains which excludes any kind of impurities or secondary phase.

In order to broaden the field of investigation, transmission electron microscopy was used for the ${\mathrm{Pr}}_{0.7}{\mathrm{Sr}}_{0.3}{\mathrm{Mn}}_{0.85}{\mathrm{Co}}_{0.15}O_{3}$ sample to highlight the nature of the particle size distribution, its average size, as well as its form via ImageJ software (version 1.46), as shown in Figure 5. The TEM micrograph indicates that the samples have nearly spherical shapes and the histogram shows a homogeneous particle size distribution estimated at 89.59 nm with a polydispersity index (PDI) of 0.37. It seems clear that all the samples studied by SEM and TEM consist of particles significantly higher than the values obtained by XRD. This is because XRD determines the coherent domain of diffraction (the crystallite size) and TEM and SEM determine the particle size. The crystallite size is always smaller than the particle size. It can be concluded that each particle is multidomain.

Semi-quantitative chemical analysis by EDS (see Figure 6), coupled with TEM indicated the presence of all the chemical elements (Pr, Sr, Mn and Co) constituting the compounds ${\;\mathrm{Pr}}_{0.7}{\mathrm{Sr}}_{0.3}{\mathrm{Mn}}_{\left(1-x\right)}{\mathrm{Co}}_{x}O_{3}$ with chemical composition close to the nominal one used in the synthesis of these compounds. The slight difference between the nominal composition and the measured composition (Table 4) indicates a good reaction of all the elements during the elaboration of these compounds by the sol-gel method.

### 3.3. Magnetic Measurements

Figure 7a shows the temperature dependence of zero field cooled magnetization (ZFC) and field cooled magnetization (FC) for ${\;\mathrm{Pr}}_{0.7}{\mathrm{Sr}}_{0.3}{\mathrm{Mn}}_{\left(1-x\right)}{\mathrm{Co}}_{x}O_{3}$ with (0 ≤ x ≤ 0.15) in a magnetic field of 500 Oe. From this figure, it is clear that all compounds show similar magnetic behavior, including a transition from the FM phase at low temperature (T < T_C_) to the PM phase above the Curie temperature T_C_, with a very clear irreversibility deduced from the difference between the ZFC-FC magnetization branches. A reasonable estimate of the Curie temperature T_C_ can be obtained from the inflection point of the magnetization derivatives dM(T)/dT, (Figure 7b). This temperature decreases from 300 K for x = 0 to 235 K for the compound with a Cobalt rate, x = 0.15. This can be interpreted by the fact that the presence of Co^3+^ in our compounds weakens the double exchange interactions in favor of the super exchange interactions between Mn/Co ions. On the other hand, the substitution of Mn ions by ions with smaller ionic radii reduces the <r_B_> (Table 2), which changes the <θ_Mn,Co-O-Mn,Co_> angles and <d_Mn,Co-O_> distances, thus decreasing the Curie temperature T_C_ [23,40]. The substitution rate x increases the disorder effect in the B site. This disorder qualifies the difference of the sizes of the cations in the B site, it reflects the local deformations of the MnO_6_ octahedra. The increase of this disorder leads to the reduction of ferromagnetism and thus to decrease the T_C_ (Table 5). Based on the double-exchange model [4], the decrease in T_C_ is strongly related to the increase in the average bond length <d_Mn,Co-O_>, which contributes to a decrease in the bandwidth W of the electrons e_g_ [23] expressed empirically by the formula W∝ cos γ/(<d_Mn-O_>)^3.5^, where γ = 1/2 (π − <θ_Mn,Co-O-Mn,Co_>) (see Table 2 and Figure 3). The decrease in W causes the reduction of the orbital overlap between the Mn-3d and O-2p orbitals, which in turn decreases the exchange coupling of Mn^3+^-Mn^4+^ and Mn^3+^-Co^3+^, and hence the decrease in the T_C_ temperature.

At a temperature T_irr_, indicated by the arrow in Figure 7a, an increasing bifurcation between the ZFC-FC curves is observed with increasing Co content, which can be explained by the presence of a small non-FM contribution probably related to the surface of the nanoparticles which, as shown by different authors, constitutes a dead layer in ferromagnetic manganite nanoparticles [41,42,43]. Moreover, T. Zhu et al. [44], using La_2/3_Sr_1/3_MnO_3_ nanoparticles, prepared by sol-gel, observed a spin-glass behavior, similar to the one observed with our nanoparticles, which is associated with the nanoparticle surface. The M-ZFC curves for the whole composition show a weak bend at low temperature. The maximum of these curves is generally defined in the manganite literature by the blocking temperature T_B_ which is a characteristic of the super paramagnetic-ferromagnetic transition [45]. This T_B_ temperature also decreases as the cobalt content increases, which is another indication of a disordered magnetic environment.

As shown in the insets of Figure 7a, we can observe a slight increase in magnetization in the ferromagnetic region (<46 K) of the unsubstituted sample (indicated by arrows in the enlarged region of the M(T) curves. Similar behavior was observed in the experimental work of Dwight et al. [46] and Hcini et al. [47]. This slight increase is probably due to the ferrimagnetic contribution of a small trace of Mn_3_O_4_ [48,49] at 46 K (it cannot be avoided by the synthesis method, and it is certainly below the detection limit of the XRD analysis in the present work). We also notice, in Figure 7b, the presence of two peaks at T = 160 K and T = 150 K for samples x = 0.05 and x = 0.15, respectively, which may be due to the particle size dispersion, because the nanopowders have nanoparticles with a different size and correspondently different Curie temperature.

The study of magnetic properties at high temperature is based on the study of the magnetic susceptibility χ vs. temperature. In Figure 8, we have plotted the variation of the inverse of the magnetic susceptibility (1/χ) vs. temperature of the ${\;\mathrm{Pr}}_{0.7}{\mathrm{Sr}}_{0.3}{\mathrm{Mn}}_{\left(1-x\right)}{\mathrm{Co}}_{x}O_{3}$ samples (x = 0, 0.05, 0.10 and 0.15).

These variations show that the susceptibility for samples with 0 ≤ x ≤ 0.15 follow a Curie–Weiss law:(4)$$\chi_{c}=T-\theta_{p}$$
for temperatures *T* > T_C_, where *C* is the Curie constant and *θ_p_* the paramagnetic Curie temperature. From the curve χ^−1^ = f (*T*), we can calculate the Curie–Weiss temperature (*θ_p_*) by extrapolating to the *x*-axis of the linear region of the paramagnetic range, and thus calculate the effective magnetic moment (μ_eff_) in Bohr magneton (μ_B_) using the relation:(5)$$\mu_{\mathrm{eff}}^{\exp\;2}=\frac{3K_{B}M_{m}C}{N_{A}\mu_{B}^{2}}$$
where K_B_ = 1.3807 × 10^−23^ J.K^−1^ is the Boltzmann constant, C = (1/slope) is the Curie constant of the linear part of the paramagnetic range, N_A_ = 6.023 × 10^23^ mol^−1^ is the Avogadro number, M_m_ is the molecular weight, and μ_B_ = 9.274 × 10^−21^ emu is the Bohr magneton.

Generally, the magnetic moment is constituted of a contribution due to the electronic spin and another due to the orbital moment and is calculated in the following way: $\mu_{\mathrm{eff}}=g_{J\;}\sqrt{J\left(J+1\right)}$. In the case of transition metals (nd layers) where the orbital angular moment is blocked, L = 0, the expression for the effective moment reduces to: $\mu_{\mathrm{eff}}=g_{J\;}\sqrt{S\left(S+1\right)}$, where g is the Landé factor and S is the cation spin. G = 2 and S = 3/2 for Mn^4+^ and 2 for Mn^3+^. Considering a rigid coupling between Mn^3+^ and Mn^4+^, the theoretical effective paramagnetic moment per unit of formula of our samples is:(6)$$\mu_{\mathrm{eff}}^{\mathrm{th}}\left(\mu_{B}\right)=\sqrt{0.7{\left(\mu_{\mathrm{eff}}^{\mathrm{th}}({\mathrm{Pr}}^{3+})\right)}^{2}+\left(0.7-x\right){\left(\mu_{\mathrm{eff}}^{\mathrm{th}}({\mathrm{Mn}}^{3+})\right)}^{2}+\;x{\left(\mu_{\mathrm{eff}}^{\mathrm{th}}({\mathrm{Co}}^{3+})\right)}^{2}+0.3{\left(\mu_{\mathrm{eff}}^{\mathrm{th}}({\mathrm{Mn}}^{4+}\right)}^{2}}$$
where: $\mu_{\mathrm{eff}}({\mathrm{Pr}}^{3+})=3.58{\;\mu }_{B}$;$\mu_{\mathrm{eff}}({\mathrm{Mn}}^{3+})\approx 4.90{\;\mu }_{B}$; $\mu_{\mathrm{eff}}({\mathrm{Mn}}^{4+})=3.87{\;\mu }_{B};{\;\mu }_{\mathrm{eff}}\left({\mathrm{Co}}^{3+}\right)\approx 4.89\mu_{B}$ are the effective spin moments (The contribution of the orbital moment is neglected).

The values of $\mu_{\mathrm{eff}\;}^{\mathrm{th}}$, $\mu_{\mathrm{eff}\;}^{\exp}$ and the Curie–Weiss $\theta_{p}$ temperature are shown in Table 5. According to the table, the Curie–Weiss temperature θ_p_ is positive and near to T_C_, decreasing with the increase of substitution rate x, indicating a strong ferromagnetic type interaction between the spins, in agreement with the mean field theory. The large difference between the experimental $\mu_{\mathrm{eff}\;}^{\mathrm{th}}$ and theoretical $\mu_{\mathrm{eff}\;}^{\exp}$ values is a sign of weakening of the DE interactions due to the cationic disorder introduced by Co^3+^ in the lattice.

To get a clearer understanding of the magnetic interaction for the ${\;\mathrm{Pr}}_{0.7}{\mathrm{Sr}}_{0.3}{\mathrm{Mn}}_{\left(1-x\right)}{\mathrm{Co}}_{x}O_{3}$ series with (0 ≤ x ≤ 0.15), the magnetization M (emu/g) was measured under the application of the magnetic field ($\mu_{0}H\;$= −5T to 5 T) at room temperature (300 K) and at low temperature (5 K), as shown in Figure 9. These curves confirm the magnetic transition of all samples from the FM phase at low temperature to the PM phase at room temperature with a slight decrease in net magnetization as the substitution rate increases, indicating that the introduction of Cobalt in the parent compound Pr_0.7_Sr_0.3_MnO_3_ leads to the weakening of the ferromagnetic double exchange due to the disorder that cobalt introduces in the lattice. As shown in Figure 9b,c and through Table 6, the hysteresis curves show that the magnetization strongly increases for low values of the applied magnetic field (μ_0_H = 1T) and then tends to saturation, characteristic of the ferromagnetic behavior. No anti-ferromagnetic contribution is observed, indicating that Co has a predominantly Co^3+^ valence. According to the magnetic phase diagram of the ${\mathrm{Pr}}_{1-x}{\mathrm{Sr}}_{x}{\mathrm{Mn}}_{x}O_{3}$ system [50], the composition of our samples is: ${\mathrm{Pr}}_{0.7}^{3+}{\mathrm{Sr}}_{0.3}^{3+}{\mathrm{Mn}}_{0.3}^{4+}{\mathrm{Mn}}_{0.7-x}^{3+}{\mathrm{Co}}_{x}^{3+}O_{3.00}^{2-}$. This result is also confirmed by the results obtained from the thermogravimetry analysis.

### 3.4. Electrical Transport Properties

Resistance measurements, with and without field, were performed following a cycle of cooling down (CD) from room temperature to 4 K and subsequent warming up (WU) to room temperature. It is important to highlight the practical absence of hysteresis in this process. The variation of resistivity with temperature was measured at different applied fields 0T and 14T, for all materials, as shown in Figure 10.

In the ρ vs. T curves at H = 0 T, three regions can be clearly seen. When cooling down, a semiconductor behavior is observed in all materials with an increase in resistivity with decreasing temperatures until it reaches a maximum. The temperature of the maximum generally decreases with increasing Co content, similar to the Curie temperature of materials. Thereafter, the materials present a metallic behavior with a decrease in resistivity with decreasing temperatures until reaching a minimum at T ≈ 30 K. Finally, the resistance increases again up to T = 4 K. This last effect can be attributed to the potential barrier between particles [40,42]. The application of magnetic field 14T produces a significant decrease in the resistivity value, which is especially pronounced in the region of the maximum. This fact translates into high magnetoresistance (MR) values (see Figure 11 and Figure 12). Two facts stand out in this graph. First, the highest MR values are obtained for the x = 0.05 material and, second, the high MR values at room temperature presented by the undoped material (45.25% for T = 300 K) and the x = 0.10 and x = 0.15 materials (~40% and 38% at T = 300 K).

From Table 7, we notice a high value of magnetoresistance at room temperature with increasing cobalt content in the PSMO system which makes this system very important for multifunctional applications at room temperature.

#### 3.4.1. Electrical Behavior at Low Temperature (80 K < T_ρ_)

In the low temperature region (T < T_ρ_), the behavior of manganites is generally metallic, which can be modeled for temperatures above 80 K as follows:(7)$$\rho =\rho_{0}+\rho_{2}T^{2}$$
(8)$$\rho =\rho_{0}+\rho_{2.5}T^{2.5}$$
(9)$$\rho =\rho_{0}+\rho_{2}T^{2}+\rho_{4.5}T^{4.5}$$
where ρ_0_ is the resistivity due to grain/domain boundary and point defects scattering [55], ρ_2_T^2^ in (7) and (9) represents the electrical resistivity due to the electron–electron scattering [56]. ρ_2.5_T^2.5^ is the electrical resistivity due to electron–magnon scattering process in the ferromagnetic phase [57], and the term ρ_4.5_T^4.5^ is a combination of electron–electron, electron–magnon, and electron–phonon scattering process [58]. To better understand the nature of the low temperature conduction mechanism, the experimental data of Pr_0.7_Sr_0.3_Mn_(1−x)_Co_x_O_3_ compounds (x = 0, 0.05, 0.10 and 0.15) were fitted using the above three equations. The curve fit ρ versus T for all samples is shown in Figure 13. The goodness of fit is generally evaluated by comparing the square of the linear correlation coefficient (R^2^) obtained for each equation. As shown in Table 8, it is clear that the obtained R^2^ values for all samples were higher than 99%, which represent the best goodness of fit.

From the Table 8, it can be seen that the parameters ρ_0_, ρ_2_, and ρ_4.5_ increase with the increase of Co rate, according to the reduction of the double exchange mechanism. The value of the electron-electron scattering term ρ_2_ is greater than that of the electron-(magnon, phonon) scattering term ρ_4.5_ for all samples. The electron-electron scattering appears to be predominant over that of the electron-(magnon, phonon) scattering in the metallic regime.

#### 3.4.2. Electrical Behavior at High Temperature (T > T_ρ_)

The variation of electrical resistivity with temperature above T_ρ_ can be explained on the basis of two distinct models in two temperature ranges: The first one is the VRH “Variable Range Hopping” model [59], a Mott model indicating a conduction by hopping of polarons at distance, considered in the range T_ρ_ < T < θ_D/2_. The other is the ASPH model “Adiabatic Small polaron Hopping” [60], a model of “hopping” of small polaron considered for T > θ_D/2_. With θ_D/2_, the Debye temperature.

#### VRH Model

The conduction can be described using the Mott variable range hopping (VRH) model:(10)$$\sigma =\sigma_{0}\exp{\left(\frac{T_{0}}{T}\right)}^{1/4}$$
where (σ = 1/ρ) represents the conductivity, and T_0_ is the Mott temperature such that:(11)$$T_{0}=\frac{16\alpha^{3}}{K_{B}N\left(E_{F}\right)}$$

The value of T_0_ is obtained from the slope of the ln(σ) vs. T^−1/4^ graphs (see Figure 14). α is a constant and was taken as 2.22 nm^−1^ [56] and N(E_F_) is the density of states at the Fermi level. The values of T_0_ and N(E_F_) are given in Table 9. These results show that with increasing Co, the T_0_ values increase and the N(E_F_) values decrease. This decrease can be attributed to the decrease of the double exchange (DE) interaction [61].

#### ASPH Model

At high temperature (T > θ_D/2_), the small polaron adiabatic hopping (ASPH) model governs the conduction mechanism of Pr_0.7_Sr_0.3_Mn_(1−x)_Co_x_O_3_ samples (0 ≤ x ≤ 0.15). This model is given by the following equation:(12)$$\rho =\mathrm{AT}\exp\left(\frac{E_{a}}{K_{B}T}\right)$$
where A is a coefficient of resistivity, and E_a_ is the activation energy for “hopping” conduction. From the slope of the graph Ln(ρ/T) vs. (1000/T) (see Figure 15), E_a_ was calculated for all samples and is listed in Table 9.

It is observed from Table 9 that the activation energy increases as the number of hole carriers increases with the increase of Co^3+^.

## 4. Conclusions

In this work, we studied the influence of Cobalt substitution, in terms of structural, magnetic, and electrical transport properties, at the Mn-site in the ${\;\mathrm{Pr}}_{0.7}{\mathrm{Sr}}_{0.3}{\mathrm{Mn}}_{\left(1-x\right)}{\mathrm{Co}}_{x}O_{3}$ (0 ≤ x ≤ 0.15) system synthesized by the Pechini Sol-gel method. Thermogravimetric analysis shows that the formation of a stable perovskite phase for all samples starts at 600 °C. The Rietveld refinement of the XRD models shows that all samples crystallized in an orthorhombic structure with the space group Pnma. The average crystallite sizes of the samples obtained by different methods are not similar due to the microstrain within the material. The morphological study showed that all the samples consist of highly packed particles that form a quasi-spherical shape distributed in a homogeneous microstructure. The M-T curves show a clear ferromagnetic (FM) paramagnetic (PM) transition and the presence of a glassy magnetic state transition at low temperature. A significant decrease in Curie temperature (T_C_) is observed and net magnetization with increasing Cobalt content due to the decrease in double exchange interactions (DE) with regard to Pr_0.7_Sr_0.3_MnO_3_. The M-H curves show a PM behavior at room temperature (300 K) while the magnetic hysteresis loops show an FM behavior at low temperature (5 K). After a detailed analysis of the electrical resistivity curves ρ(T), we find that our Pr_0.7_Sr_0.3_Mn_(1−x)_Co_x_O_3_ (0 ≤ x ≤ 0.15) samples show a metallic behavior at low temperature (80 K < T_ρ_) well fitted by the relation ρ = ρ_0_+ ρ_2_T^2^ + ρ_4.5_T^4.5^ and a semiconductor behavior above T_ρ_ (T > T_ρ_) well fitted by both variable range hopping and adiabatic small polaron hopping models. All the doped materials show high MR values that enhance those of the undoped sample.

## Figures and Tables

**Figure 1 materials-16-01573-f001:**
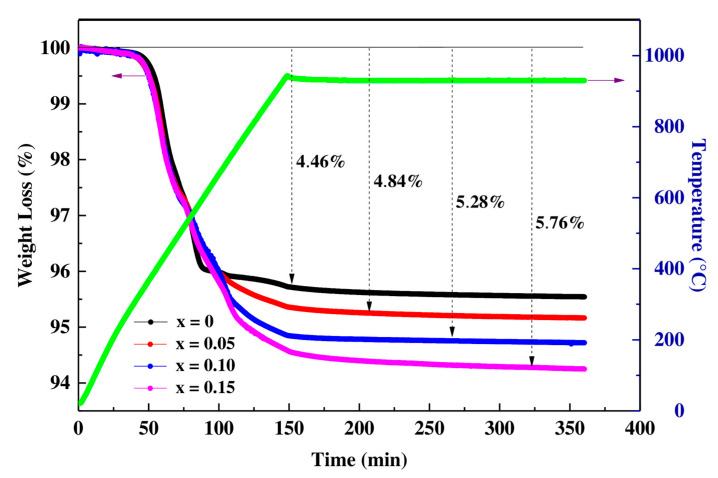
Thermogravimetric analysis of ${\mathrm{Pr}}_{0.7}{\mathrm{Sr}}_{0.3}{\mathrm{Mn}}_{\left(1-x\right)}{\mathrm{Co}}_{x}O_{3\;}$ samples with (0 ≤ x ≤ 0.15) versus the temperature and time.

**Figure 2 materials-16-01573-f002:**
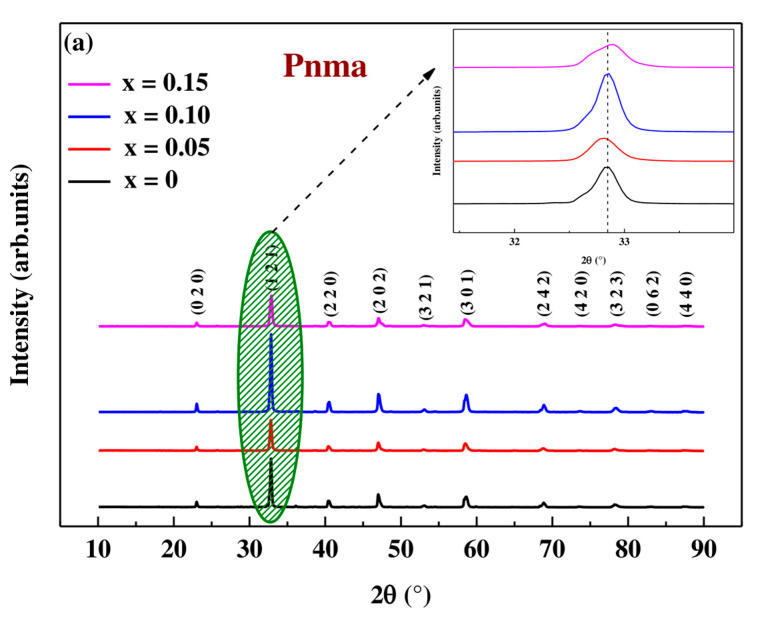
(**a**) X-ray diffraction patterns of ${\mathrm{Pr}}_{0.7}{\mathrm{Sr}}_{0.3}{\mathrm{Mn}}_{\left(1-x\right)}{\mathrm{Co}}_{x}O_{3}$ (x = 0, 0.05, 0.10 and 0.15) compounds at room temperature. (**b**) Profile of the measured (Open circles) and calculated (Strong lines) intensity of the X-ray diffraction lines obtained by the Rietveld method. The vertical bars in green are Bragg’s reflections for the Pnma space group. The difference between the observed and calculated intensity is shown in blue at the bottom.

**Figure 3 materials-16-01573-f003:**
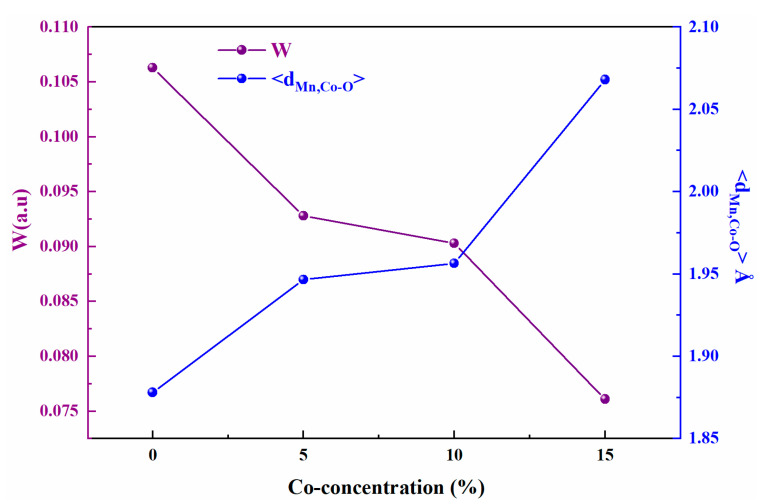
Variation of the bandwidth W and interatomic distance as a function of the Cobalt rate for ${\mathrm{Pr}}_{0.7}{\mathrm{Sr}}_{0.3}{\mathrm{Mn}}_{\left(1-x\right)}{\mathrm{Co}}_{x}O_{3}$ (0 ≤ x ≤ 0.15).

**Figure 4 materials-16-01573-f004:**
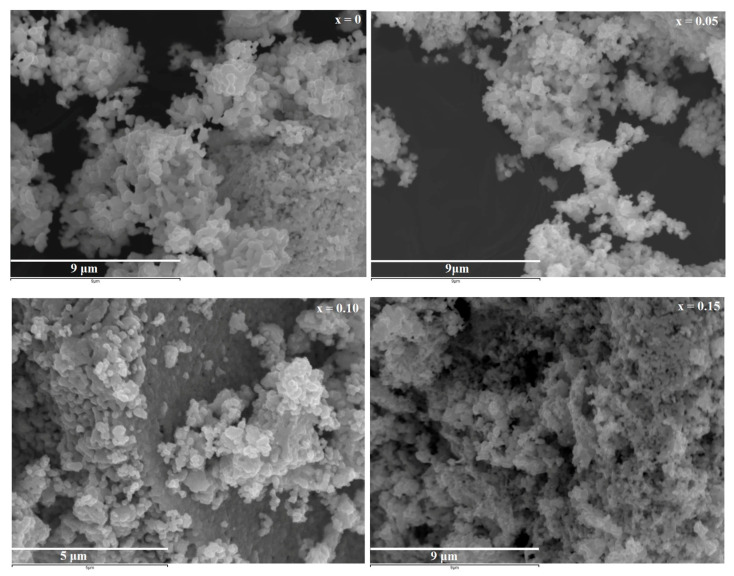
SEM Micrographs of ${\;\mathrm{Pr}}_{0.7}{\mathrm{Sr}}_{0.3}{\mathrm{Mn}}_{\left(1-x\right)}{\mathrm{Co}}_{x}O_{3}$ (0 ≤ x ≤ 0.15) samples.

**Figure 5 materials-16-01573-f005:**
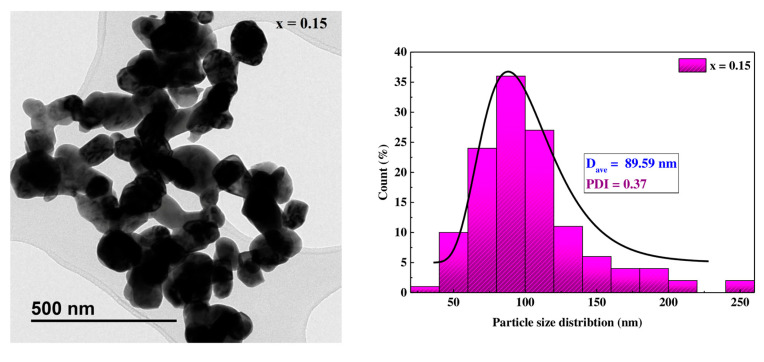
TEM image and size distribution fitted to a log normal function of nanocrystaline ${\mathrm{Pr}}_{0.7}{\mathrm{Sr}}_{0.3}{\mathrm{Mn}}_{0.85}{\mathrm{Co}}_{0.15}O_{3}$ sample.

**Figure 6 materials-16-01573-f006:**
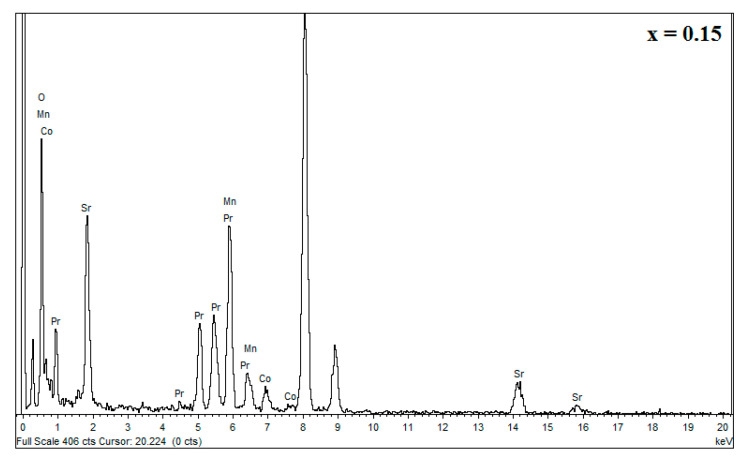
EDS spectrum showing the different chemical elements in the ${\mathrm{Pr}}_{0.7}{\mathrm{Sr}}_{0.3}{\mathrm{Mn}}_{0.85}{\mathrm{Co}}_{0.15}O_{3}$.

**Figure 7 materials-16-01573-f007:**
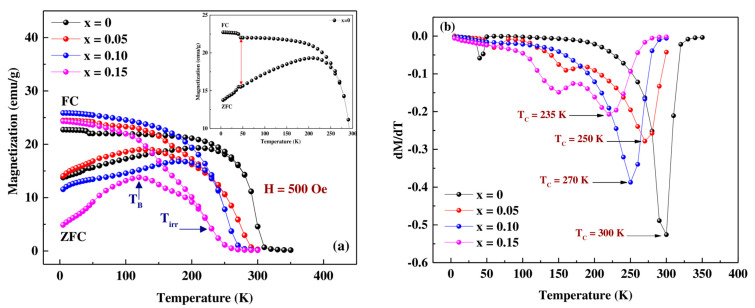
Magnetization versus temperature of ${\;\mathrm{Pr}}_{0.7}{\mathrm{Sr}}_{0.3}{\mathrm{Mn}}_{\left(1-x\right)}{\mathrm{Co}}_{x}O_{3}$ (x = 0, 0.05, 0.10 and 0.15): (**a**). Temperature dependence of zero-field cooled (ZFC) and field cooled (FC) magnetization curves measured at 500 Oe showing a net PM-FM phase transition at Curie temperature (T_C_) estimated in (**b**) by determining the minimum value of the dM/dT curves versus T. T_irr_ (indicated by an arrow) shows the irreversibility and drop in magnetization in ZFC mode. T_B_ is the blocking temperature.

**Figure 8 materials-16-01573-f008:**
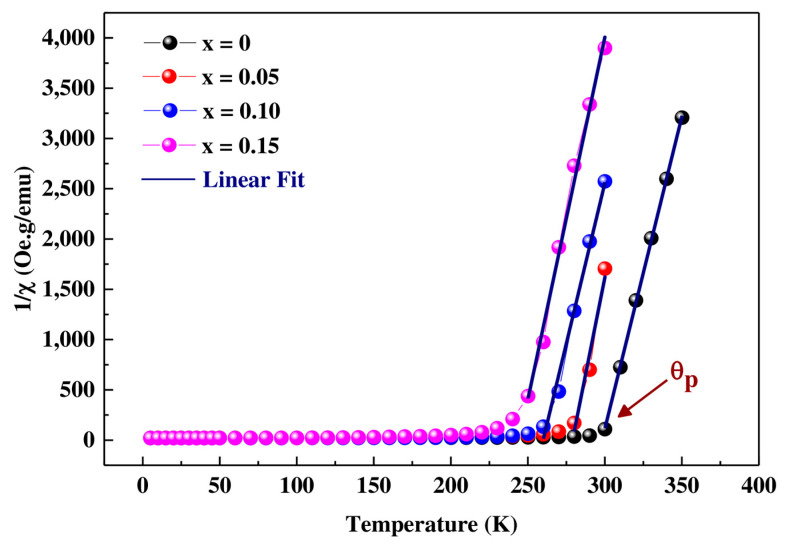
Inverse of magnetic susceptibility versus temperature for the compounds ${\;\mathrm{Pr}}_{0.7}{\mathrm{Sr}}_{0.3}{\mathrm{Mn}}_{\left(1-x\right)}{\mathrm{Co}}_{x}O_{3}$ (0 ≤ x ≤ 0.15).

**Figure 9 materials-16-01573-f009:**
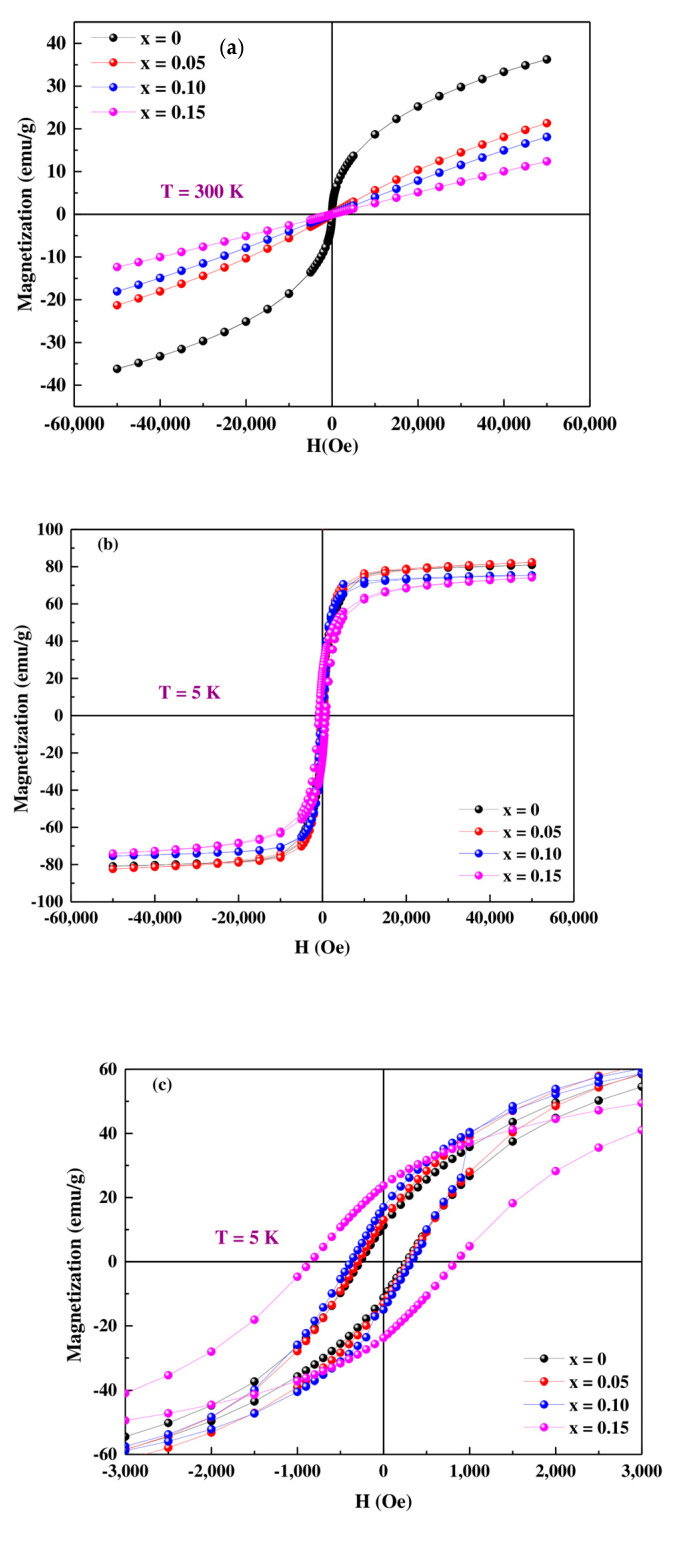
Magnetization curves versus applied magnetic field of ${\mathrm{Pr}}_{0.7}{\mathrm{Sr}}_{0.3}{\mathrm{Mn}}_{\left(1-x\right)}{\mathrm{Co}}_{x}O_{3}$ nanocrystalline samples (0 ≤ x ≤ 0.15): (**a**). At 300 K (**b**) at 5 K (**c**). Enlarged region of the M(H) curves at 5 K.

**Figure 10 materials-16-01573-f010:**
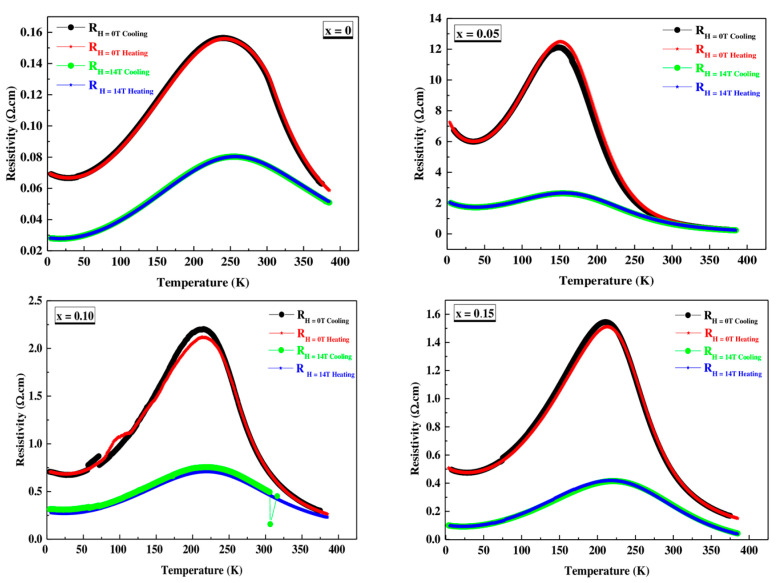
Temperature dependence of the resistivity ρ(T,H) without and with applied magnetic field for ${\;\mathrm{Pr}}_{0.7}{\mathrm{Sr}}_{0.3}{\mathrm{Mn}}_{\left(1-x\right)}{\mathrm{Co}}_{x}O_{3}$ with x = 0, 0.05, 0.10 and 0.15.

**Figure 11 materials-16-01573-f011:**
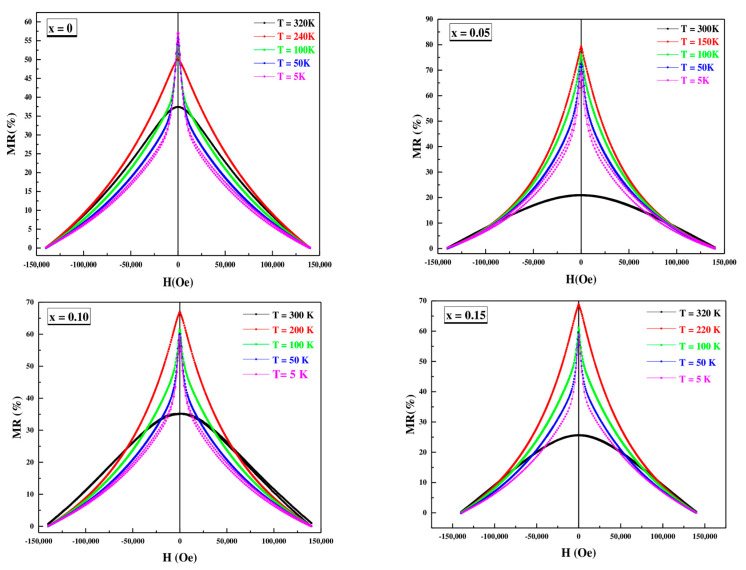
Magnetoresistance (%MR) vs. applied field at different temperature for ${\;\mathrm{Pr}}_{0.7}{\mathrm{Sr}}_{0.3}{\mathrm{Mn}}_{\left(1-x\right)}{\mathrm{Co}}_{x}O_{3}$ nanopowders.

**Figure 12 materials-16-01573-f012:**
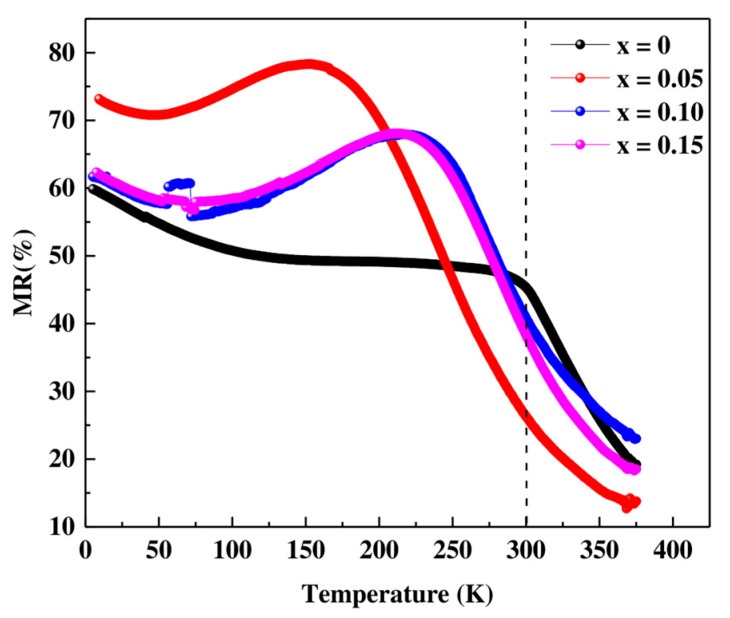
Variation of Magnetoresistance (%MR) vs. T, at different Co contents for ${\;\mathrm{Pr}}_{0.7}{\mathrm{Sr}}_{0.3}{\mathrm{Mn}}_{\left(1-x\right)}{\mathrm{Co}}_{x}O_{3}$.

**Figure 13 materials-16-01573-f013:**
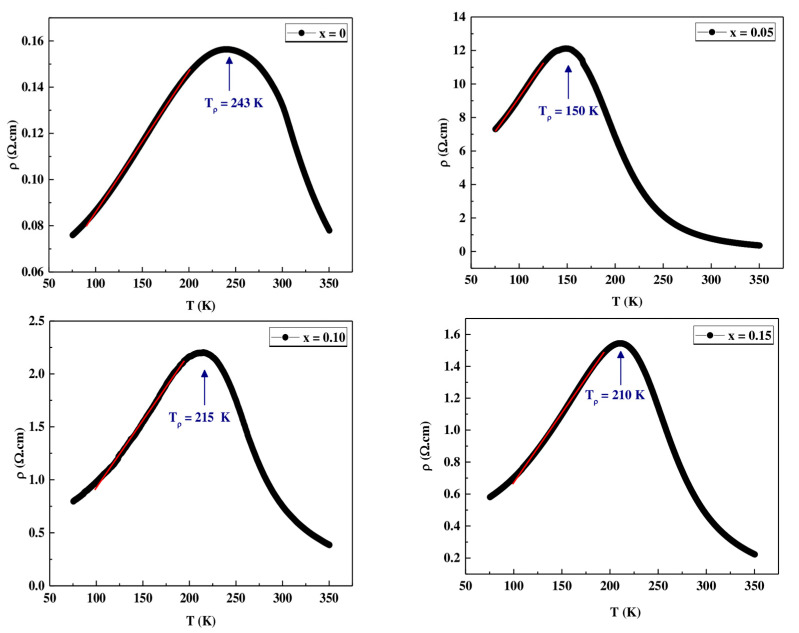
Plots of the electrical resistivity ρ(T) vs. T, for ${\;\mathrm{Pr}}_{0.7}{\mathrm{Sr}}_{0.3}{\mathrm{Mn}}_{\left(1-x\right)}{\mathrm{Co}}_{x}O_{3}$ (x = 0, 0.05, 0.10 and 0.15). The red line corresponds to fit by with Equation (9).

**Figure 14 materials-16-01573-f014:**
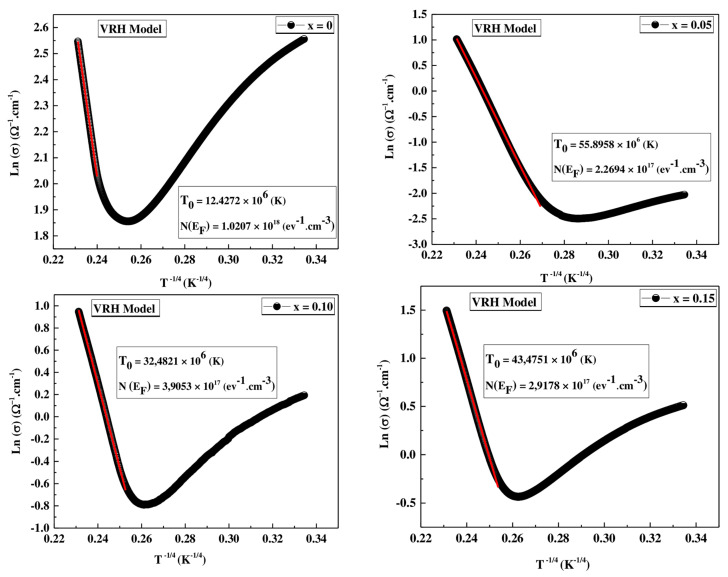
Plots of ln(σ) vs. T^−1/4^ for ${\;\mathrm{Pr}}_{0.7}{\mathrm{Sr}}_{0.3}{\mathrm{Mn}}_{\left(1-x\right)}{\mathrm{Co}}_{x}O_{3}$ (0 ≤ x ≤ 0.15) compounds. The red solid line corresponds to fit by Equation (10).

**Figure 15 materials-16-01573-f015:**
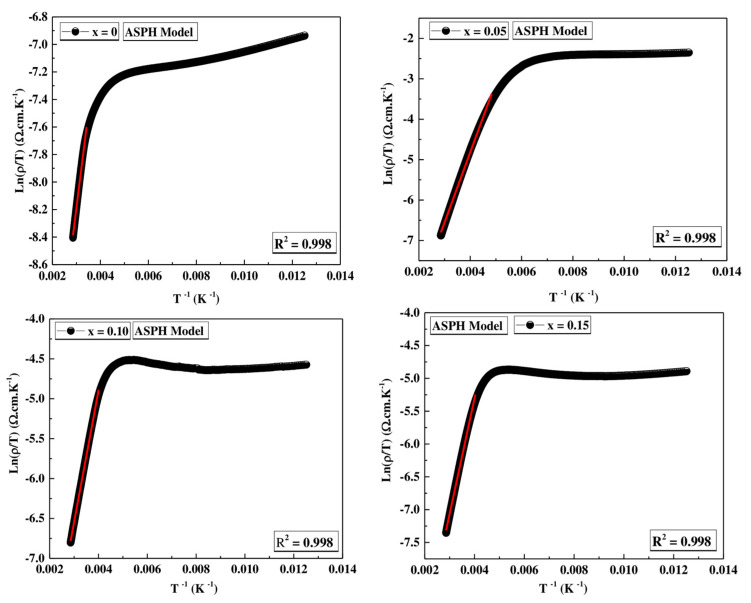
Plots of Ln (ρ/T) vs. inverse temperature for ${\;\mathrm{Pr}}_{0.7}{\mathrm{Sr}}_{0.3}{\mathrm{Mn}}_{\left(1-x\right)}{\mathrm{Co}}_{x}O_{3}$ (0 ≤ x ≤ 0.15) compounds. The red solid line corresponds to fit by Equation (12).

**Table 1 materials-16-01573-t001:** Results of thermogravimetric analysis for ${\mathrm{Pr}}_{0.7}{\mathrm{Sr}}_{0.3}{\mathrm{Mn}}_{\left(1-x\right)}{\mathrm{Co}}_{x}O_{y}$ (x = 0, 0.05, 0.10 and 0.15). The errors are included in parenthesis.

	Initial Mass (mg)	Final Mass (mg)	Oxygen Rate (y)	Compositions
x = 0	47.67 (1)	45.54 (1)	2.98 (1)	${\mathrm{Pr}}_{0.7}{\mathrm{Sr}}_{0.3}{\mathrm{MnO}}_{2.98\;}$
x = 0.05	72.64 (1)	69.13 (1)	2.99 (1)	${\mathrm{Pr}}_{0.7}{\mathrm{Sr}}_{0.3}{\mathrm{Mn}}_{0.95}{\mathrm{Co}}_{0.05}O_{2.99\;}$
x = 0.10	58.31 (1)	55.23 (1)	3.00 (1)	${\mathrm{Pr}}_{0.7}{\mathrm{Sr}}_{0.3}{\mathrm{Mn}}_{0.90}{\mathrm{Co}}_{0.10}O_{3.00\;}$
x = 0.15	55.59 (1)	52.39 (1)	3.01 (1)	${\mathrm{Pr}}_{0.7}{\mathrm{Sr}}_{0.3}{\mathrm{Mn}}_{0.85}{\mathrm{Co}}_{0.15}O_{3.01\;}$

**Table 2 materials-16-01573-t002:** Structural parameters refined by the Rietveld method of compounds ${\mathrm{Pr}}_{0.7}{\mathrm{Sr}}_{0.3}{\mathrm{Mn}}_{\left(1-x\right)}{\mathrm{Co}}_{x}O_{3}$ (0 ≤ x ≤ 0.15) at room temperature.

x	0	0.05	0.10	0.15
Space group	Pnma			
Lattice parameter				
a (Å)	5.4494 (28)	5.4426 (0)	5.4465 (16)	5.4590 (0)
b (Å)	7.6913 (43)	7.7355 (0)	7.6921 (24)	7.6651 (0)
c (Å)	5.4811 (27)	5.4715 (0)	5.4787 (16)	5.4697 (0)
V (Å^3^)	229.738 (0)	230.359 (0)	229.533 (0)	228.878 (0)
W (u.a)	0.1092	0.0960	0.0941	0.0779
t_G_	0.9198	0.9206	0.9214	0.9222
<r_B_> (Å)	0.6105	0.6087	0.6070	0.6052
d_Mn,Co-O1_ (Å)	1.956 (4)	1.94685 (0)	1.9540 (17)	1.94540 (0)
θ_Mn,Co-O1-Mn,Co_ (°)	159 (10)	166.7649 (0)	159.6 (6)	160.1462(0)
d_Mn,Co-O2_ (Å)	1.80 (4)	1.94644 (0)	1.959 (14)	2.1907 (0)
θ_Mn,Co-O2-Mn,Co_ (°)	170.9 (15)	158.9012 (0)	162.6 (7)	170.8186 (0)
<d_Mn,Co-O_> (Å)	1.878	1.9466	1.9565	2.068
<θ_Mn,Co-O-Mn,Co_> (°)	164.95	162.833	161.1	165.4824
Discrepancy factors (%)				
R_p_ (%)	12.7	17.5	7.46	16.5
R_wp_ (%)	13.6	17	6.68	15
R_F_ (%)	4.03	4.27	3.37	4.01
χ^2^ (%)	9.57	10.2	3.5	8.75

**Table 3 materials-16-01573-t003:** Crystallite size (from XRD) of the ${\mathrm{Pr}}_{0.7}{\mathrm{Sr}}_{0.3}{\mathrm{Mn}}_{\left(1-x\right)}{\mathrm{Co}}_{x}O_{3}$.

x	0	0.05	0.10	0.15
D_SCH_ (nm)	33.78	29.00	33.61	24.27

**Table 4 materials-16-01573-t004:** EDS chemical analysis for ${\mathrm{Pr}}_{0.7}{\mathrm{Sr}}_{0.3}{\mathrm{Mn}}_{0.85}{\mathrm{Co}}_{0.15}O_{3}$ sample.

Sample $Pr_{0.7}Sr_{0.3}Mn_{\left(1-x\right)}Co_{x}O_{3}$	Element	Weight (%)	Nominal Chemical Composition	Experimental Chemical Composition
x = 0.15	Mn	27.98	0.85	0.861
	Co	4.49	0.15	0.138
	Sr	24.90	0.30	0.368
	Pr	42.63	0.70	0.630

**Table 5 materials-16-01573-t005:** Values of the Blocking temperature T_B_, the Irreversibility temperature T_irr_, the Curie temperature T_C_, the Curie constant C, the Curie–Weiss temperature θ_p_ and the experimental and theoretical effective paramagnetic moment for ${\;\mathrm{Pr}}_{0.7}{\mathrm{Sr}}_{0.3}{\mathrm{Mn}}_{\left(1-x\right)}{\mathrm{Co}}_{x}O_{3}$ (0 ≤ x ≤ 0.15).

Composition Pr_0.7_Sr_0.3_Mn_(1−x)_Co_x_O_3_	T_B_ (K)	T_irr_ (K)	T_C_ (K)	θ_p_ (K)	C (emu mol^−1^ Oe^−1^ K)	$\mu_{eff}^{th}$ $\left(\mu_{B}\right)$	$\mu_{eff}^{exp}$ $\left(\mu_{B}\right)$
x = 0	220	229	300	295	0.0161	5.5019	3.5899
x = 0.05	139	190	270	275	0.0129	5.5015	3.2239
x = 0.10	190	210	250	255	0.0156	5.5010	3.5417
x = 0.15	119	180	235	238	0.0138	5.5006	3.3310

**Table 6 materials-16-01573-t006:** Values of saturated magnetization (Ms), coercivity Hc and residual magnetization Mr, at 5 K and 300 K for samples ${\mathrm{Pr}}_{0.7}{\mathrm{Sr}}_{0.3}{\mathrm{Mn}}_{\left(1-x\right)}{\mathrm{Co}}_{x}O_{3}$ (0 ≤ x ≤ 0.15).

Composition Pr_0.7_Sr_0.3_Mn_(1−x)_Co_x_O_3_	Hc (Oe)	Mr (emu/g)	Ms (emu/g)	Hc (Oe)	Mr (emu/g)	Ms (emu/g)
	5 K			300 K		
x = 0	272.13	11.14	80.92	8.76	0.157	36.24
x = 0.05	299.01	13.06	82.40	12.89	0.007	21.32
x = 0.10	377.28	16.86	75.10	12.64	0.005	18.00
x = 0.15	852.36	23.70	74.13	12.42	0.003	12.17

**Table 7 materials-16-01573-t007:** MR values (%) at room temperature and in a magnetic field H = 10 kOe for ${\;\mathrm{Pr}}_{0.7}{\mathrm{Sr}}_{0.3}{\mathrm{Mn}}_{\left(1-x\right)}{\mathrm{Co}}_{x}O_{3}$ samples (x = 0, 0.05; 0.10 and 0.15) compared to several systems.

Compositions	T_C_ (K)	Tρ (K)	MR (%)			Ref
			T (K)	H (kOe)	%	
Pr_0.7_Sr_0.3_MnO_3_	300	243	320	10	35.9	This work
Pr_0.7_Sr_0.3_Mn_0.95_Co_0.05_O_3_	270	150	300	10	20.5	This work
Pr_0.7_Sr_0.3_Mn_0.90_Co_0.10_O_3_	250	215	300	10	34.5	This work
Pr_0.7_Sr_0.3_Mn_0.85_Co_0.15_O_3_	235	210	320	10	25.3	This work
La_0.7_Ca_0.3_Mn_0.9_Co_0.1_O_3_	188	155	170	4	14.3	[51]
Pr_0.7_Sr_0.3_MnO_3_	-	-	175	10	47.1	[52]
La_0.67_Sr_0.33_Mn_0.90_Co_0.10_O_3_	294	250	300	50	21	[26]
Sm_0.7_Sr_0.3_MnO_3_	-	-	77	30	90	[53]
La_0.7_Sr_0.3_MnO_3_	369	>360	78	5	15	[54]

**Table 8 materials-16-01573-t008:** Fit parameters to Equation (9) obtained for the metallic behavior (below Tρ ) for ${\;\mathrm{Pr}}_{0.7}{\mathrm{Sr}}_{0.3}{\mathrm{Mn}}_{\left(1-x\right)}{\mathrm{Co}}_{x}O_{3}$ samples (0 ≤ x ≤ 0.15) and Metal-semiconductor transition temperature Tρ.

$\rho_{FM}\left(T\right)=\rho_{0}+\rho_{2}T^{2}+\rho_{4.5}T^{4.5}$							
	ρ_0_ (Ω.cm)	ρ_2_ (Ω.cm.K^−2^)	ρ_4.5_ (Ω.cm.K^−4.5^)	ρ_max_ (Ω.cm)	R^2^	T_ρ_ (K)	ρ_0_ (Ω.cm)
x = 0	0.0061	9.1812 × 10^−8^	6.1581 × 10^−12^	0.1560	0.996	243	0.0061
x = 0.05	0.0785	7.5708 × 10^−6^	5.6774 × 10^−10^	12.0645	0.997	150	0.0785
x = 0.10	0.0124	3.0995 × 10^−7^	6.4824 × 10^−11^	2.2014	0.992	215	0.0124
x = 0.15	0.0085	2.2599 × 10^−7^	5.0214 × 10^−11^	1.5447	0.996	210	0.0085

**Table 9 materials-16-01573-t009:** Values of activation energy Ea, correlation factors R and Mott temperature (T_0_) for ${\;\mathrm{Pr}}_{0.7}{\mathrm{Sr}}_{0.3}{\mathrm{Mn}}_{\left(1-x\right)}{\mathrm{Co}}_{x}O_{3}$ (0 ≤ x ≤0.15) using VRH (Equation (10)) and ASPH (Equation (12)) models.

Models	VRH Model $\rho =\rho_{0}exp{\left(-\frac{T_{0}}{T}\right)}^{1/4}$			ASPH Model $\rho_{PM}=ATexp\left(-\frac{E_{a}}{K_{B}T}\right)$	
	T_0_ (10^6^) (K)	N(E_F_) (ev^−1^.cm^−3^)	R^2^	E_a_ (meV)	R^2^
x = 0	12.4272	1.0207 × 10^18^	0.999	107.51	0.998
x = 0.05	55.8958	2.2694 × 10^17^	0.998	153.84	0.998
x = 0.10	32.4821	3.9053 × 10^17^	0.999	140.57	0.998
x = 0.15	43.4751	2.9178 × 10^17^	0.998	153.87	0.998

## Data Availability

Not applicable.

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
