# Peer review of "Effects of Partial Manganese Substitution by Cobalt on the Physical Properties of Pr0.7Sr0.3Mn(1−x)CoxO3 (0 ≤ x ≤ 0.15) Manganites"

_materials, 2023, doi:10.3390/ma16041573_

Round 1
Reviewer 1 Report
Zdiri et al. have presented the manuscript titled: Effects of partial manganese substitution by Cobalt on the physical properties of Pr0.7Sr0.3Mn(1-x)CoxO3 (0 ≤ x ≤ 0.15) manganites. Overall presentation of the article is good, and authors have provided the detailed study. I just have few suggestions for the authors about this article.
1. Last sentence of the abstract please relate the material under study with the practical application or devices.
2. There is a big family of materials for perovskite manganites, where scientists have explored their magnetic and dielectric properties. I think it will be better to provide the literature review mentioning the results (values) of those reported samples for comparison with this study.
In fact authors can prepare the table for such studies regarding their properties.
3. What standard structure authors have used to measure the refinement.A
Author Response
The answers are in the attachment.

Reviewer 2 Report
The paper is focused on obtaining manganite nanopowders Pr0.7Sr0.3Mn1-xCoxO3 (0≤x≤0.15) and studying their structural, magnetic, and transport properties in wide temperature and magnetic field ranges. The authors use many different experimental and theoretical approaches, including XRD, SEM, EDS, TEM, TGA, magnetic, resistance, magnetoresistance, and VRH, ASPH. They defined a lot of essential parameters, which, however, need to be clarified. From the practical application point of view, the high values of the MR effect have been obtained for the studied samples, which makes this work worthwhile. In general, the manuscript corresponds to the aims and scope of the Materials and may interest readers. Therefore, I recommend accepting this manuscript after major revision(s).
1. Since the paper is devoted to studying the influence of substitution, oxygen stoichiometry, defect chemistry, distortions, particle size, inhomogeneities, etc., on the physical properties of the manganites, I recommend supplementing the "Introduction" with some related and relatively new research in this direction (https://doi.org/10.1016/j.jmmm.2015.06.042, https://doi.org/10.1016/j.ceramint.2021.05.174). Moreover, the "Introduction" should be decreased in length, and general principles do not need to be described as DE.
2. From Table 1, it is hard to understand why with a reduction in sample mass after sintering, the oxygen content increases but not decreases? And how extra oxygen more than "O3" can exist in a perovskite structure?
3. Double-check the oxygen radius in the manuscript (0.61 Å). It seems wrong (http://abulafia.mt.ic.ac.uk/shannon/radius.php?Element=O). Furthermore, check the tolerance factor values in Table 2. Was the oxygen deficiency used during the calculation of the tolerance factor? Why only Co3+ in HS was used for consideration of structural properties, but not Co3+ in LS or Co4+ in HS replacing Mn4+? Probably, additional XPS measurements need to be carried out.
4. It was noted that " We also notice in Figure.7(b), the presence of two peaks at T = 160 K and T = 150 K for samples x = 0.05 and x = 0.15 respectively, which could be due to the high sintering temperature that affects the quality of the sample. ", but in my opinion, it may be associated with a particle size dispersion, i.e. nanopowders have nanoparticles with a different size and correspondently different Curie temperature.
5. Eq. (6) is wrong since Pr3+ does not have a magnetic moment with a closed electronic configuration; for example, Pr = 4f3 6S2. So, an effective magnetic moment should be recalculated.
6. Please, also define saturated magnetization, coercivity, and residual magnetization.
7. Electrical behavior should also be discussed at lower temperatures < 30 K. Additionally, the Kondo-like spin scattering ρSln(T) and the electron-electron interaction ρelT1/2 contributions should be addressed. The recommended literature is https://doi.org/10.1016/j.apmt.2021.101340.
8. There needs to be a discussion about the obtained results (activation energy, etc.) after using the ASPH model.
9. MR data should supplement conclusions for the studied samples to highlight the value of this manuscript.
10. General remarks:
All abbreviations have to be checked (FM, PM, EDS, SEM, TEM, DE, etc.) since they are created to be abbreviated again and again.
Check "(a). At 300 K (b) at 5K (c). Enlarged region of the M(H) curves at 5K.", p. 14
Check "of the resistivity ρ(T,H) at without and with applied", p. 15
"As shown in Table.2, it is clear that the obtained R2 values" should be Table 6, p. 17
"The values of T0 and N(EF) are given in Table.6." should be Table 7, p. 18
Author Response
The answers are in the attachment

Reviewer 3 Report
In this work, the authors have investigated the effect of substituting different concentrations of Mn for Co in the compound Pr_0.7Sr_0.3Mn(_1-x)Co_xO_3. Samples were fabricated in powder by sol-gel method and characterized using different techniques. Then, the authors have studied the Co substitution effects on the magnetic and electrical properties of the samples. From the technical point of view, the work is scientifically sound and seems technically correct. From a general point of view, I think that this work has potential to be published in this journal after the following minor changes:
è The presentation of the manuscript needs a thorough revision to avoid typos like, for example, between lines 77 and 78 “the magnetic transition from the ferromagnetic state (FM) to the ferromagnetic state (PM)”. Moreover, acronyms FM and PM are also multiple defined. In fact, the surname of Author 6 seems written in a wrong way.
On the other hand, there are several understandable sentences. See, for instance, the idea between lines 78-81, starting with “On the other hand,..”, which is not clear. The authors also mentioned a contrast: what contrast? Why that contrast?
è In line 78 the authors defined x as a rate, “rate of substitution x increases”. Is x a rate or a ratio/percentage?
è The reference section also missed some recent references which the authors should cite and mention in the introduction section:
o https://doi.org/10.3390/ma12060861
o https://doi.org/10.1016/j.ceramint.2017.03.063
Author Response
The answers are in the attachment

Round 2
Reviewer 2 Report
Formally, the authors' reply needs to be completed and detailed. However, further clarification of the current manuscript does not lead to significant improvement, so that the current version may be accepted.
Author Response
The answers are in the attached file.
